# Optimization of Selective Laser Melting Parameter for Invar Material by Using JAYA Algorithm: Comparison with TLBO, GA and JAYA

**DOI:** 10.3390/ma15228092

**Published:** 2022-11-15

**Authors:** Hiren Gajera, Faramarz Djavanroodi, Soni Kumari, Kumar Abhishek, Din Bandhu, Kuldeep K. Saxena, Mahmoud Ebrahimi, Chander Prakash, Dharam Buddhi

**Affiliations:** 1Department of Mechanical Engineering, L D College of Engineering, Ahmedabad 380015, India; 2Department of Mechanical Engineering, College of Engineering, Prince Mohammad Bin Fahd University, Al Khobar 31952, Saudi Arabia; 3Department of Mechanical Engineering, Imperial College London, London SW7 2AZ, UK; 4Department of Mechanical Engineering, GLA University, Mathura 281406, India; 5Department of Mechanical and Aero-Space Engineering, Institute of Infrastructure, Technology, Research and Management (IITRAM), Ahmedabad 380026, India; 6Department of Mechanical Engineering, Indian Institute of Information Technology Design and Manufacturing (IIITDM), Kurnool 518008, India; 7National Engineering Research Center of Light Alloy Net Forming and Key State Laboratory of Metal Matrix Composites, School of Materials Science and Engineering, Shanghai Jiao Tong University, Shanghai 200240, China; 8Division of Research and Development, Lovely Professional University, Phagwara 144011, India; 9Division of Research & Innovation, Uttaranchal University, Dehradun 248007, India

**Keywords:** DMLS, sintering, ANOVA, taguchi, invar, hardness, surface roughness, JAYA, TLBO, GA

## Abstract

In this study, the hardness and surface roughness of selective laser-melted parts have been evaluated by considering a wide variety of input parameters. The Invar-36 has been considered a workpiece material that is mainly used in the aerospace industry for making parts as well as widely used in bimetallic thermostats. It is the mechanical properties and metallurgical properties of parts that drive the final product’s quality in today’s competitive marketplace. The study aims to examine how laser power, scanning speed, and orientation influence fabricated specimens. Using ANOVA, the established models were tested and the parameters were evaluated for their significance in predicting response. In the next step, the fuzzy-based JAYA algorithm has been implemented to determine which parameter is optimal in the proposed study. In addition, the optimal parametric combination obtained by the JAYA algorithm was compared with the optimal parametric combination obtained by TLBO and genetic algorithm (GA) to establish the effectiveness of the JAYA algorithm. Based on the results, an orientation of 90°, 136 KW of laser power, and 650 mm/s scanning speed were found to be the best combination of process parameters for generating the desired hardness and roughness for the Invar-36 material.

## 1. Introduction

The concept of Rapid Prototyping (RP) is basically the fabrication of products or models using three-dimensional computer-aided design (CAD). This is an ‘additive’ process, in which layers of paper, wax, plastic, metal, ceramics, concrete, etc. are fused together to form the final object. Most traditional processes, such as milling, drilling, grinding, etc., are subtractive in nature, which means that material is removed from a solid block. Rapid Prototyping’s additive nature allows it to create objects with complicated internal features that cannot be manufactured by other conventional methods. In addition, RP techniques are widely popular in the aerospace, automobile, and tooling sector [1,2,3,4,5]. Among various RP technologies, the Powder-Based Fusion (PBF) or Selective Laser Melting (SLM) process is widely popular to manufacture metal parts by the principle of rapid prototyping technique. Nowadays, sectors such as aerospace and automobile have been trying to supply accurate parts in the minimum time; quality and cost-effective parts must be delivered and orders fulfilled in this cut-throat industry. [6,7].

Invar-36 belongs to the category of iron-nickel alloys. In earlier days, Invar-36 specimens were manufactured from a bulky wrought workpiece which led to high cost. Moreover, industries have been facing problems in fabricating complex specimens of Invar-36 material [8]. The Invar material is widely popular for its low thermal expansion rate at room temperature. Hence, Invar material is used to manufacture equipment that required high thermal stability and precision positioning under varying temperature conditions [9,10]. Due to this capacity of Invar, it is widely used in the aerospace and automobile sector where the material of parts can sustain their thermal stability at high temperatures. The complex fuel injector nozzle has been assembled using multiple parts. Additionally, it required too much time and energy to make one fuel injector nozzle. So, these problems can be avoided by the PBF process which has eliminated all of these issues as a result of the aforementioned steps and fabrication process of the final product. Hence, the Invar material is a suitable material for the as-mentioned sectors for producing complex geometry of products or parts [1,9,10,11,12].

Despite having too many unique characteristics of the SLM method, it has been facing certain limitations like surface finish, mechanical properties, dimensional accuracy, etc. In that regard, Yakout et al. conducted a review of the various SLM methods. They have concluded that still little research has been explored to find the optimal parameter for the density and mechanical properties of SLM-made products [6]. Harrison et al. compared the mechanical properties of the SLM Invar specimen with the cold-drawn Invar-36 material. They observed that the low thermal expansion property of Invar material is not affected by the SLM method. It further reduced the thermal expansion coefficient for atmospheric temperature [13]. Qiu et al. conducted a study to examine the effect of scanning speed and orientation on the tensile behavior and density of SLM-made specimens. Afterward, they measured the effect of heat treatment on the same response. They revealed that the porosity and the cracking problem have been raised by increasing the scanning speed at constant laser power. The study also revealed that vertically orientated specimens are affected by the pores. Moreover, the effect of heat treatment on the tensile strength was adverse. They concluded that the low coefficient of thermal expansion up to 300 °C was not affected by said process [14]. Yakout et al. conducted a study to find the optimal parameter for the Invar material. They fabricated the specimen of Invar material by using various combinations of the process parameters, and they compared this study with the maraging steel as a competitor material. They concluded that the density of the part has been raised with the rising energy density. They suggested that the dense specimen of Invar and maraging steel can be fabricated at 60 to 75 J/mm^3^ and 67.5 J/mm^3^ of laser energy density, respectively. They said that Invar materials have required a high amount of energy density while having low thermal conductivity [15]. Yakout et al. conducted a study to examine the thermal expansion coefficient of Invar material and stainless-steel-316. They suggested that laser energy densities affect significant parameters for the density and mechanical properties. For that, they suggested that the laser energy density must be higher than the critical energy to fabricate the dense specimen. The high amount of energy density provides a high amount of coefficient of thermal expansion. However, in contradiction, the high amount of laser energy density evaporates the reduction in the manganese, chromium, and nickel concentrations. Hence, the thermal expansion coefficient has been reduced [16]. The cost of the surface finish process can be reduced by reducing the surface roughness of the parts. It can be possible only by arranging such process parameters that minimize surface roughness. The surface roughness, hardness, and density of parts must be required to avoid the huge loss of defective parts as well as human loss in any industry [17].

Hence, the aforementioned responses must be attained to fulfill the requirement of industries. From the past literature, it can be said that surface roughness and hardness are the significant parameters for the PBF-made specimen. All these studies provide significant information on the density and tensile behavior of SLM-made specimens. However, it does exclude any discussion for surface roughness and hardness of SLM made as well as the effect of laser power, scanning speed, and orientation on the Invar material. Therefore, comprehensive research is imperative in that direction. This work aims to establish the optimal parametric set by using the desirability function approach and explore key process parameters that affect the aforementioned responses.

## 2. Experimentation

The Invar material is very reliable and precise along with having a low thermal expansion coefficient. Due to these characteristics, Invar material has been used in vital structures like gas pipelines, storage tanks, and ships [12]. Spectro analysis was used to estimate the chemical composition of Invar as shown in Table 1 (which is very similar to that of the actual Invar material). To manufacture specimens (Figure 1), the SLM technique was applied to the M1 Cusing model (fabricated by the Concept Laser Company (Lichtenfels, Germany)).

With a build volume of 250 mm × 250 mm × 250 mm (x, y, z), SLM uses a fiber laser to melt powder materials into solid specimens. In the working chamber, nitrogen was used to protect the environment while maintaining an oxygen concentration of 1.8%. A 200 W laser was used, with a scanning speed of 7 m/s, a laser beam diameter of 0.03 mm, and a layer thickness of 0.02 to 0.08 mm.

In this present study, the surface roughness and hardness of SLM fabricated specimens have been examined, and after that heat treatment has been applied to enhance the aforementioned responses. During the heat treatment process, specimens were treated under an induction furnace at a temperature of 650 °C with a holding time of 2 h. The cooling air was selected as the quenching media. The hardness and surface roughness were measured by the hardness analyzer (Model: VRS-150) and surface roughness tester Model; SJ210 (Make: Mitutoyo, Japan), respectively. For hardness, preload have given 10 kg which is called zero position, and the 150 kg loaded Rockwell C scale (HRC) which is the major load, was applied to the specimen by the application of a diamond indenter. Experimental runs have been performed as per Taguchi’s L_8_ orthogonal array design by manipulating the process parameters, viz. laser power, scanning speed, and orientation to examine the output parameters such as hardness and surface roughness.

It is a requirement to determine the energy density to select the appropriate factors and their acceptable range of values. In this regard, the maximum energy density determined the laser power and scanning speed. The relative laser energy per unit area linked to the powder bed surface, which may be determined using Equation (1), characterizes energy density. To enhance energy density, larger laser power values are often used for producing porosity-free metal products. In this manner, the laser power is adjusted at 125 watts. To investigate the interaction impact, the laser strength is reduced to 110 and 120 watts. For the Invar metal, machine makers propose a laser power range of 90 to 130 watts. Yet, appropriate joining did not take place at 90 and 100 watts of laser power. Additionally, at more than 140 watts of laser power, the material burned and turned into an extremely rigid surface. Furthermore, the balling effect was spotted in the preliminary trial when the laser power of 140 watts was supplied to the fabricated specimens. Hence, in this investigation, laser power ranged from 115 to 136 watts, which is ideal for Invar metal. According to the machine maker’s recommendations, the scanning speed for Invar should be 600 mm/s. An increase in scanning speed increases productivity, whereas a decrease in scanning speed causes a rise in energy density.

The maximum energy density governed the laser power, scanning speed, and orientation. During the fabrication of the specimen, 0.015 mm of hatch spacing and 0.03 mm of layer thickness were held constant. The relative laser energy per unit area linked to the powder bed surface is termed the energy density, which could be determined using Equation (1) [18,19].
(1)$$\mathrm{Energy density}\;\left(\mathrm{ED}\right)=\frac{Laser\;power\;\left(watt\right)}{Hatch\;spacing\;\left(mm\right)\times Scanning\;speed\left(\frac{mm}{s}\right)}$$

Higher laser power ratings are typically employed to gain maximum ED while producing porosity-free metal products [18]. This entire study has been incorporated into three phases. In the first phase, the experimental runs have been performed to examine the aforementioned response. In the second phase, a fuzzy inference system (FIS) has been applied to translate the aforesaid response into a single response which is called the multi-performance characteristic index (MPCI) [20,21,22]. In the third phase, the regression fitness function was generated for the MPCI to run by the JAYA algorithm; and compared to the obtained result with the teaching-learning-based optimization (TLBO), genetic algorithm (GA) [23,24,25].

## 3. Proposed Optimization Methods

### 3.1. Fuzzy Inference System

A combination of a knowledge base, a fuzzifier, an inference engine, and a defuzzifier comprise a fuzzy rule-based system. Generally, the experimental results are fuzzified by assigning the suitable membership function. So, the fuzzifier converts the crisp value into imprecise information. For that, it needs to formulate the IF and THEN rule matrix which is inserted into the knowledge base fuzzy logic [21,22]. The fuzziness has performed the rule assessment throughout the intermediate step, and it delivers the final single value, which is designated an output factor. Since a fuzzy set’s integration contains a range of output values, it needs to be defuzzified to determine a single output value from the set. The centroid computation, which yields the center of the area beneath the curve, is the best frequently employed defuzzification approach. The center of gravity approach is a well-known and efficient approach for de-fuzzing fuzzy functions. The crisp value for the end good or service was estimated during the defuzzification phase employing the formula of the center of gravity (Equation (2)) [21,22].
(2)$$\hat{y_{i}}=\frac{\int y_{i}\mu_{ci}(y_{i})dy}{\int \mu_{ci}(y_{i})dy}$$
where *y_i_* indicates the sample element, *µ_ci_y_i_* is the membership function

### 3.2. Optimization Using the Jaya Algorithm

The JAYA algorithm, with the primary goal of avoiding the worst outcome, was initially envisioned by Rao [26,27,28]. It is an algorithm-specific parameter-less method. The JAYA algorithm has the propensity to strive for the ideal (success) while avoiding the worse (failure). This inclination represents the Sanskrit word JAYA, which symbolizes triumph [29]. Figure 2 depicts the flow diagram of the Jaya algorithm [26,27,28,29,30].

### 3.3. Teaching Learning-Based Optimization (TLBO)

Every evolutionary and swarm intellectual ability optimization strategy utilizes identical control factors, such as the size of the population, generation number, elite size, and so on. Aside from the identical control factor, each technique involves its algorithm-specific factors; for example, the ABC algorithm entails the bee’s number (scout, onlooker, and employed) and a limit, whereas the NSGA-2 algorithm entails mutation and crossover probability and distribution index. Appropriate adjustment of these algorithm-specific factors is a critical aspect affecting the algorithm’s effectiveness. Incorrect tweaking of algorithm-specific factors sometimes accelerates processing work and results in a local optimum response. In addition to tweaking algorithm-specific factors, identical control factors must also be modified, thereby accelerating the work. As a result, there was a need for the development of an algorithm that does not seek any algorithm-specific factors, and TLBO is one such algorithm [23].

### 3.4. Normalizing the Experimental Results

The raw data is initially processed to standardize it for assessment. Normalization is a treatment that is done on a single data value to uniformly disseminate and measure it into an appropriate limit for subsequent assessment. In the normalized data, surface roughness corresponds to the smaller-the-better (SB) criteria, which may be stated mathematically via Equation (3).
(3)$$\mathrm{Xi}(k)=\frac{\max\;\mathrm{yi}\left(k\right)\;–\;\mathrm{yi}\left(k\right)}{\max\mathrm{yi}\left(k\right)-\min\mathrm{yi}\;\left(k\right)}$$
where Xi(k) is the value for the SB criteria.

Similarly, the normalized hardness corresponds to the higher-the-better (HB) criterion which can be expressed as per Equation (4).
(4)$$\mathrm{Xi}(k)=\frac{\;\mathrm{yi}\left(k\right)-\;\min\;\mathrm{yi}\left(k\right)\;}{\max\mathrm{yi}\left(k\right)-\min\mathrm{yi}\;\left(k\right)}$$
where Xi(k) is the value for the HB criteria. Min yi(k) is the smallest value of yi(k) and for the kth response, and max yi(k) is the largest value of yi(k) for the kth response. An ideal sequence is x0(k) (k = 1, 2, …, m) for the responses.

### 3.5. Individual Optimization for Performance Characteristics

In this section, the individual optimization for each performance characteristic namely surface roughness and hardness (as furnished in Table 2) has been carried out. Individual optimization aims to target the specific goal of industries. Generally, there are several purposes to fabricating the parts by additive manufacturing; i.e., some industries are using RP technologies to make a pattern. In this case, the strength has become their specific goal which should be raised. Here, industries are only focusing on hardness or other mechanical properties. In some cases, RP-made parts are used to give a tactile feeling to parts. So, surface roughness must be less required in parts. Here, the individual goal can be varied from industry to industry according to requirements. Hence, it is very difficult to identify the process parameter where surface roughness and hardness are getting the desired result. Hence, it is essential to identify the individual optimization for the aforementioned response. In addition, the effect of input parameters on response has been identified by statistical analysis through Design expert software.

Hardness:295.9 − 1.672 × x(1) − 3.342 × x(2) − 0.09000 × x(3) + 0.01531 × x(2) × x(2) + 0.002667 × x(1) × x(3)(5)

Surface roughness:6.311 + 0.002270 × x(1) + 0.04398 × x(2) − 0.005171 × x(3) − 0.000306 × x(2) × x(2) + 0.000095 × x(1) × x(2) − 0.000022 × x(1) × x(3) + 0.000057 × x(2) × x(3)(6)

In this research, in order to conduct statistical analysis, a 95% confidence level has been taken. Generally, the value of R-square is utilized to determine how well the model can predict the response. Hence, 0% of the value of the R-square shows that the model does not change the data or there is no variance; whereas, 100% of the value of the R-square represents the capacity of the model that has higher accuracy in predicting the response. Sometimes the value of the R-square has been misleading to the experimenters. For this reason, the experimenter has accounted for the value of R-square Adj. and the value of R-square pred. For this research, it has been found that the value of R-square for hardness and surface roughness are 98.8% and 96.31, respectively. Hence, from the R-square value, it can be said that the said experimental results are accurate. It has generated the fitness function for both aforesaid responses from the experimental data as shown in Table 2. Here, the fitness function of hardness and surface roughness has been calculated using Equations (5) and (6), respectively.

To accomplish the single objective optimization, the aforementioned fitness functions of hardness and surface roughness have been run in TLBO as shown in Figure 3a and Figure 4a. For that, the lower and upper limit values of the input parameters have been considered as per Table 3. Furthermore, the same fitness function for hardness and surface roughness was run in the JAYA algorithm and GA as shown in Figure 3b,c and Figure 4b,c respectively. From Figure 5, it can be said that TLBO and JAYA algorithms have been given the final result after making 8 to 10 iterations only, whereas, the GA technique has been given the final result after 45 to 50 iterations. A similar trend has been found in the surface roughness as shown in Figure 6. In addition to this, it can also be said that the optimum value of hardness given by all algorithms is very close to each other. In the case of hardness, the value is around 71.60 Hv; whereas the value of surface roughness is around 8.12 µm.

According to the results, high laser power creates a significant energy density, that allows for the quick melting of metal powders, culminating in minuscule surface roughness. As a result, increased laser power correlates with relatively low surface roughness. Although faster scan speeds allow for less chance to melt and fuse the particulates, the surface finish of the object is compromised as a result. It is also discovered that the laser should indeed be provided with sufficient opportunity to melt and fuse particulates, culminating in a sluggish cooling process. As a result, a slower scan speed produces a superior surface finish.

Profound hardness was achieved with a laser power of approximately 115 watts and a scan speed of approximately 650 mm/s. This is owing to the premise that elevated laser power produces tremendous energy, which may expeditiously melt particulates at significant temperatures, resulting in a densely homogeneous microstructure attributable to the fast-cooling mechanism. Narrow layer thickness and elevated laser power provide superior toughness because the elevated laser power generates a homogeneous composition without any voids inside the scanned layer. Moreover, a narrow layer thickness value results in a relatively thin layer that readily melts and creates a crystalline structure with reduced porosity.

### 3.6. Optimization for Multi-Performance Characteristic Index (MPCI)

In this section, multi-objective optimization has been discussed. For that, the value of hardness and surface roughness have been converted into desirability values. Table 4 represents the individual desirability values for both responses as per the formula mentioned in Section 3.4. The higher-the-better criterion has been taken for hardness, and the lower-the-better criterion has been taken for surface roughness. In this study, all responses have been converted into desirability values as per the theory explained in Section 3.4. After that, the values have been converted into a single response by applying the concepts of FIS in Matlab 9.8 software (as explained in Section 3.1). This single response is recognized as the multi-performance characteristic index (MPCI). In this study, there is no need to assign weight by the judgment of the engineer as the FIS takes care of that.

The membership function of the input factors (responses) in the FIS is n-hardness (normalized hardness) and n-surface roughness (normalized surface roughness), as shown in Figure 7 and Figure 8, respectively. Three fuzzy sets (L, M, and H) were allocated to each input factor as depicted in Figure 7 and Figure 8. Meanwhile, five fuzzy sets (VL, L, M, H, VH) have been allocated to the outcome (MPCI) as illustrated in Figure 9. The input factors were fuzzified into the appropriate linguistic phrase, and nine logic rules have been placed in the FIS (Table 5) as depicted in Figure 10. According to the prior explanation, the MPCI must be adjusted to the maximum in order to improve the reaction. The crisp value was then calculated as the final MPCI’s outcome using the defuzzification technique by the center of gravity. In order to assess the MPCI value, each conceivable set of input factors must be specified.

Then, a mathematical model has been formulated for MPCI in terms of process parameters as shown in Equation (7).
MPCI: 19.84 − 0.09269 × x(1) − 0.2398 × x(2) − 0.01166 × x(3) + 0.000954 × x(2) × x(2) − 0.000116 × x(1) × x(2) + 0.000170 × x(1) × x(3) + 0.000037 × x(2) × x(3)(7)

Finally, this mathematical model has been implemented as a fitness function for the aforementioned algorithms. The convergence graph for the TLBO, JAYA, and Genetic algorithm has been presented in Figure 11 and the comparative convergence plot has been plotted in Figure 12. Also, the optimum result of each algorithm has been presented in Table 6.

## 4. Conclusions

In this research work, the optimal parametric set has been found for the individual response, namely hardness and surface roughness, to fulfill the specific goal of industries.
The study developed the mathematical model for the hardness and surface roughness using regression analysis. It has been noticed that it has been found that the value of R-square for hardness and surface roughness are 98.8% and 96.31%, respectively.The study utilizes different metaheuristics algorithms such as JAYA, TLBO, and genetic algorithms in order to achieve the optimal parametric combination. The optimal parametric combination for hardness has an orientation of 90°, laser power of 136 KW, and scanning speed of 650 mm/s, and surface roughness orientation of 90°, laser power of 115 KW, and scanning speed of 650 mm/s which is the same for all the above-mentioned optimization techniques.In addition, the combined goal can be achieved through only multi-objective optimization as industries have contradictory goals. In order to satisfy the contradictory goal, responses have been converted to MPCI through FIS. It has been found that an orientation of 90°, laser power of 136 KW, and scanning speed of 650 mm/s is the optimum process parameter combination to attain desired hardness and surface roughness for the Invar-36 material.Moreover, the JAYA technique is taken less time (a smaller number of iterations) to give the final results. In addition, the balling phenomena and residual stress of the SLM-made specimen can be reduced in a future study with the said parametric set.

## Figures and Tables

**Figure 1 materials-15-08092-f001:**
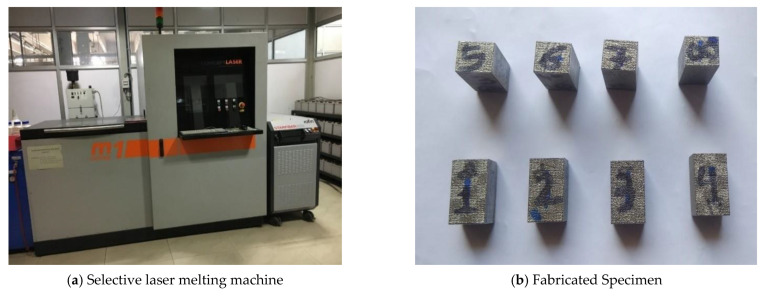
Powder-based fusion process machine.

**Figure 2 materials-15-08092-f002:**
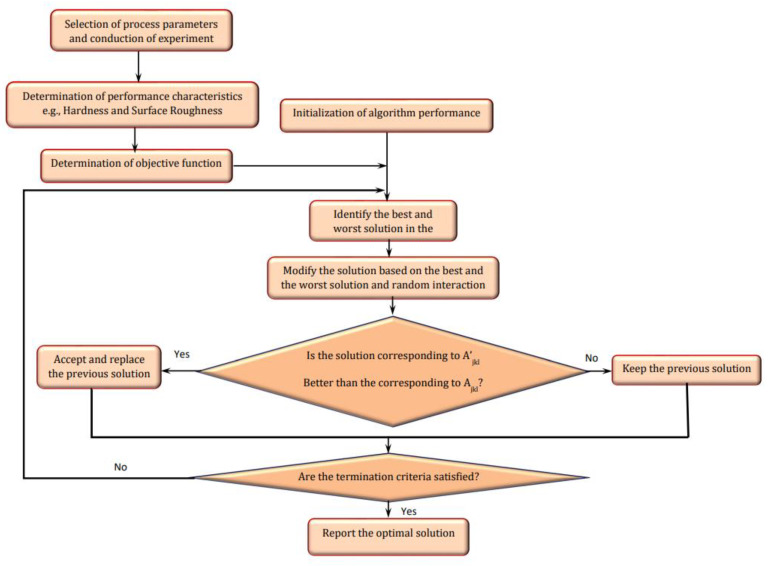
Flow chart of JAYA algorithm [26,27,28,29,30].

**Figure 3 materials-15-08092-f003:**
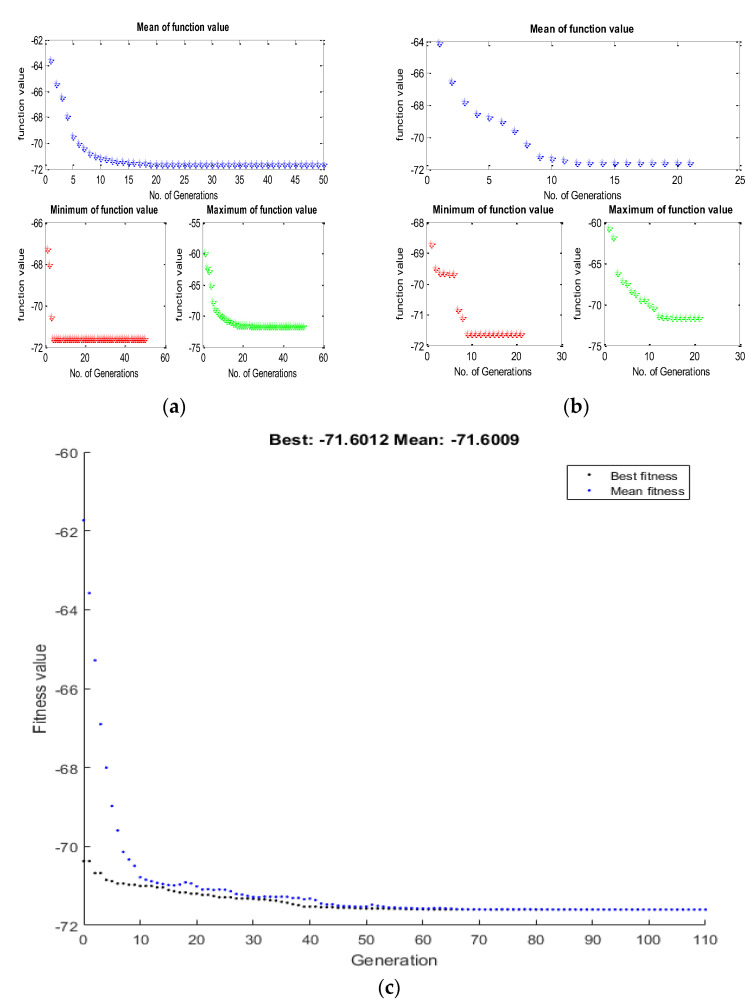
Convergence plot for the hardness by (**a**) TLBO (**b**) JAYA & (**c**) Genetic Algorithm.

**Figure 4 materials-15-08092-f004:**
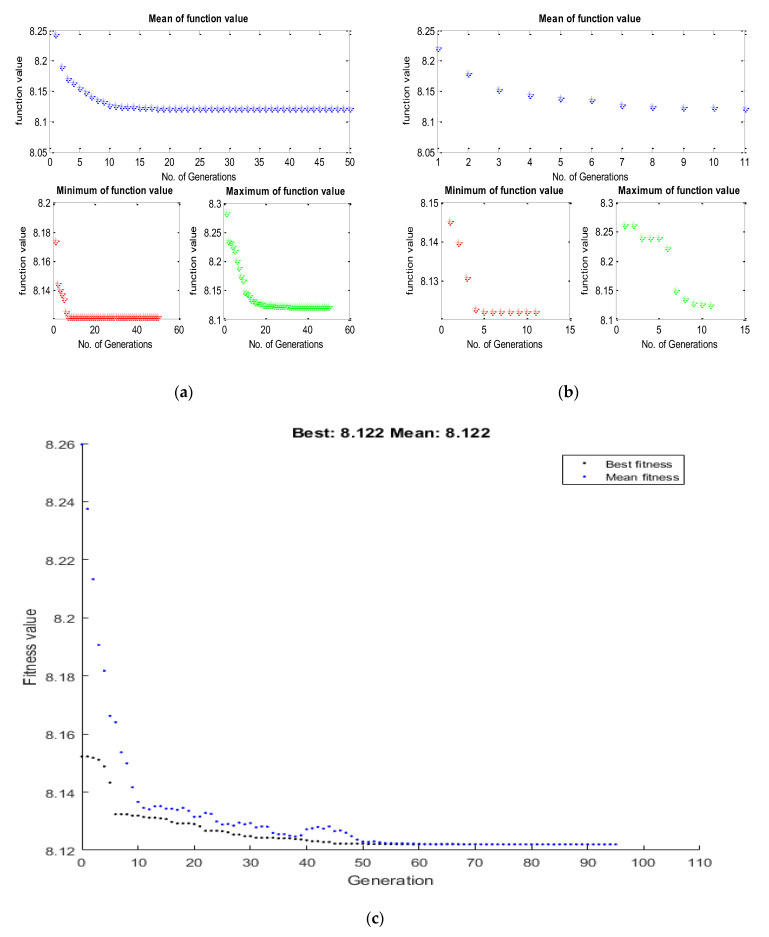
Convergence plot for the surface roughness by (**a**) TLBO (**b**) JAYA & (**c**) Genetic Algorithm.

**Figure 5 materials-15-08092-f005:**
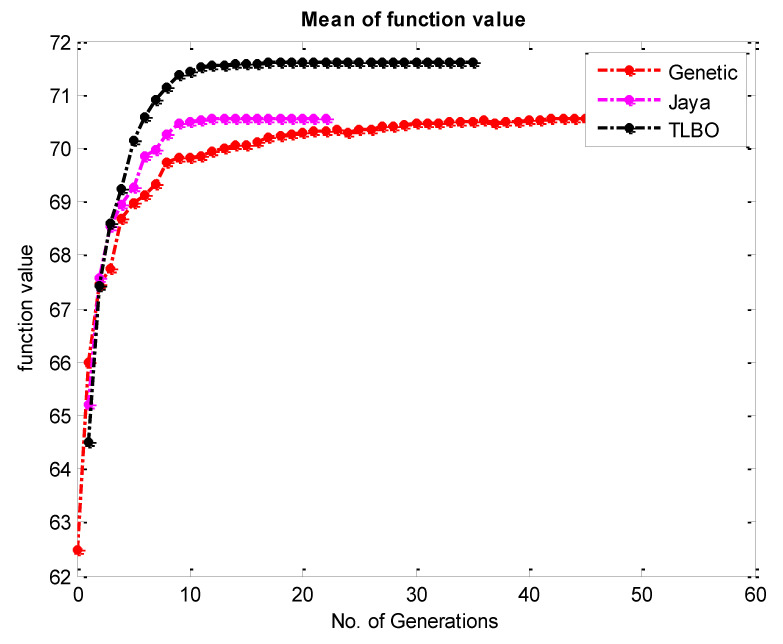
Comparative Convergence plot for the hardness by TLBO JAYA and Genetic Algorithm.

**Figure 6 materials-15-08092-f006:**
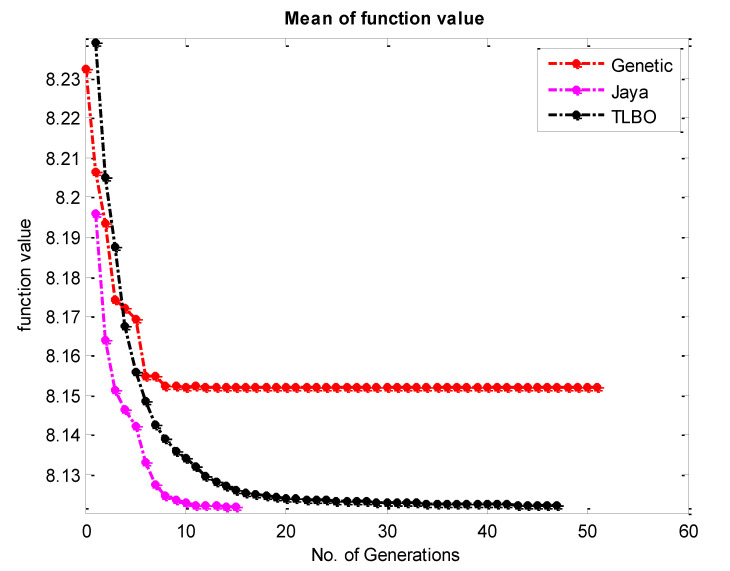
Comparative Convergence plot for the surface roughness by TLBO JAYA and Genetic Algorithm.

**Figure 7 materials-15-08092-f007:**
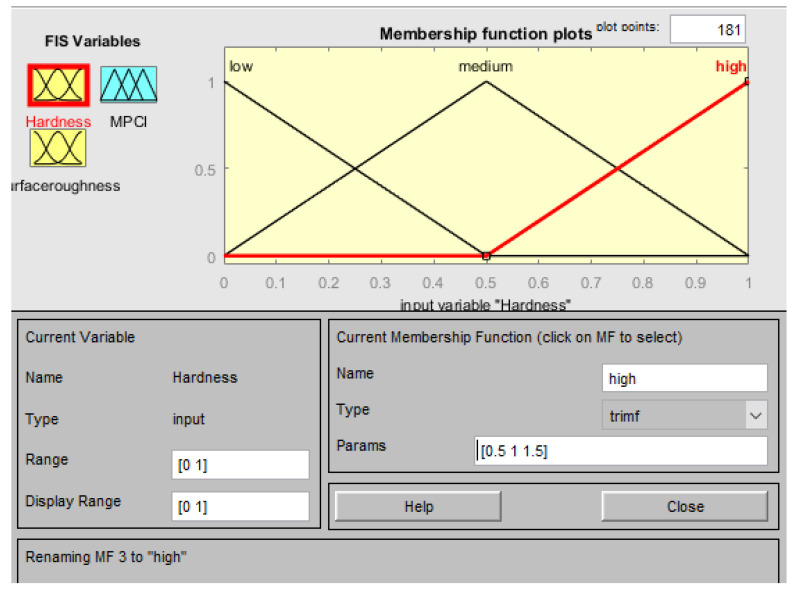
Membership function for the hardness.

**Figure 8 materials-15-08092-f008:**
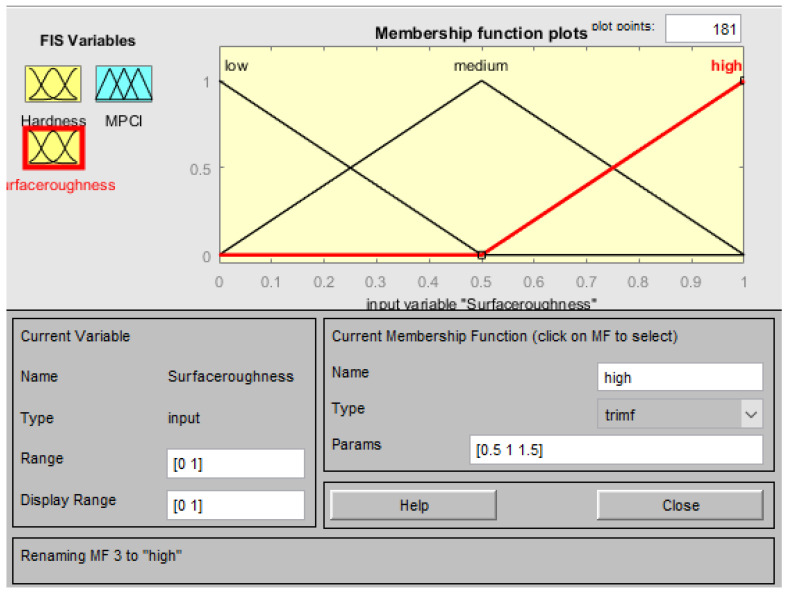
Membership function for the surface roughness.

**Figure 9 materials-15-08092-f009:**
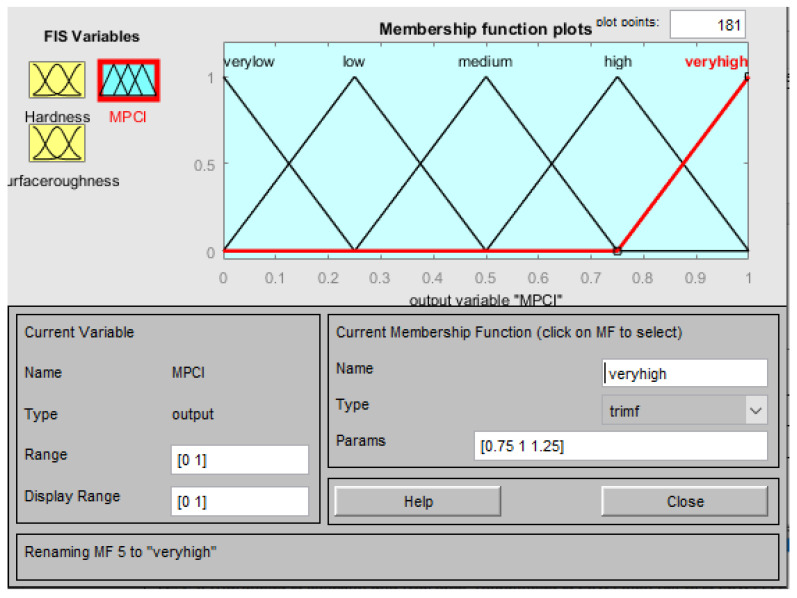
Membership function for the MPCI.

**Figure 10 materials-15-08092-f010:**
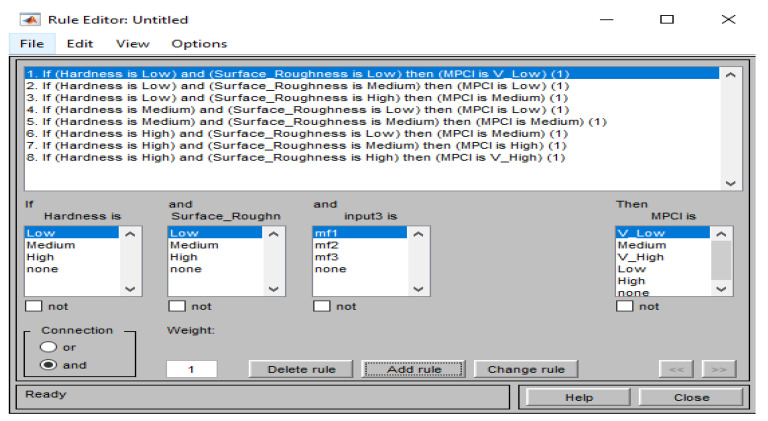
Rules editor.

**Figure 11 materials-15-08092-f011:**
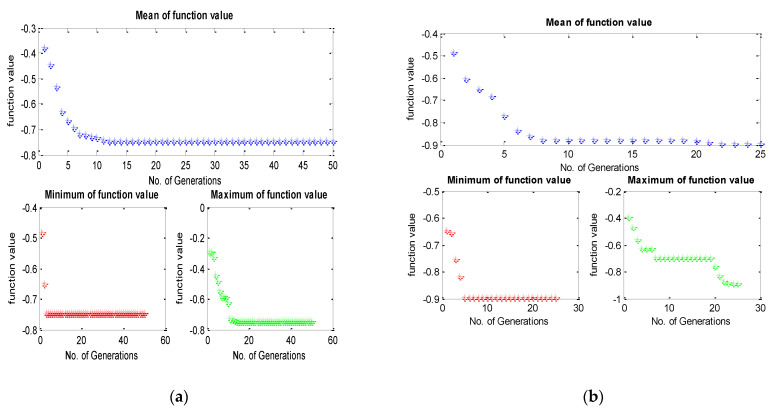
Convergence plot for the MPCI by (**a**) TLBO (**b**) JAYA & (**c**) Genetic Algorithm.

**Figure 12 materials-15-08092-f012:**
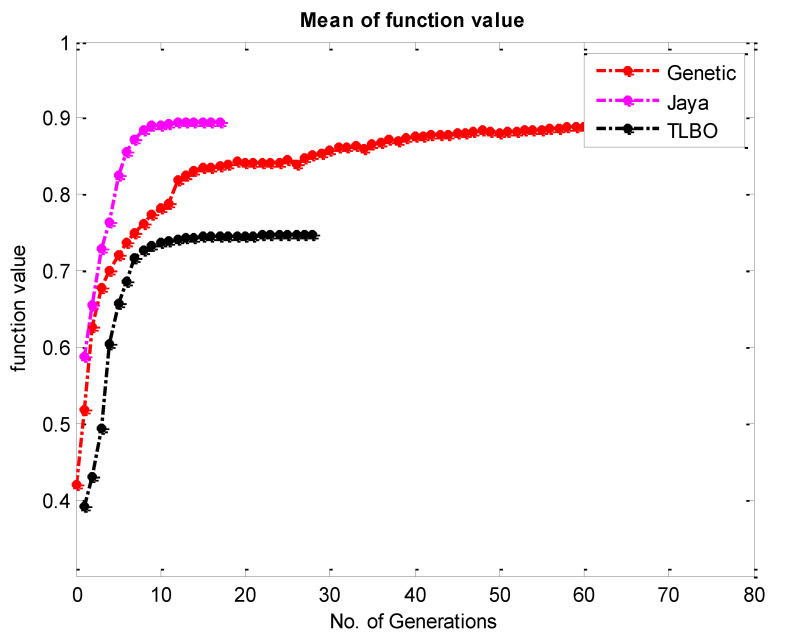
Comparative Convergence plot for the MPCI by TLBO JAYA and Genetic Algorithm.

**Table 1 materials-15-08092-t001:** Combination of chemical elements (wt.%) in Invar.

Element	C	S	P	Mn	Cr	Ni
wt.%	0.10	0.025	0.025	0.50	0.25	35.8

**Table 2 materials-15-08092-t002:** L_8_ Orthogonal array.

Sr. No.	Orientation (Degree)	Laser Power (Watt)	Scanning Speed (mm/s)	Hardness before (HRB)	Surface Roughness (µm)
1	0	115	600	60	8.16
2	90	115	650	61	8.12
3	0	122	600	62	8.20
4	90	122	650	63	8.24
5	90	129	600	59	8.32
6	0	129	650	61	8.32
7	90	136	600	64	8.36
8	0	136	650	66	8.32

**Table 3 materials-15-08092-t003:** Levels of process parameters.

Sr. No.	Process Parameter	Unit	Level 1	Level 2	Level 3	Level 4
1	Laser power	Watt	115	122	129	136
2	Scanning speed	mm/s	600	650	-	-
3	Orientation	Degree	0	90	-	-

**Table 4 materials-15-08092-t004:** L_8_ Normalizing value for hardness and surface roughness.

Sr. No.	Orientation (Degree)	Laser Power (Watt)	Scanning Speed (mm/s)	Hardness before (HRB)	Surface Roughness (µm)	MPCI
				**Normalization Value**		
1	0	115	600	0.142857	0.833333	0.44
2	90	115	650	0.285714	1	0.5
3	0	122	600	0.428571	0.666667	0.5
4	90	122	650	0.571429	0.5	0.5
5	90	129	600	0	0.166667	0.17
6	0	129	650	0.285714	0.166667	0.31
7	90	136	600	0.714286	0	0.344
8	0	136	650	1	0.166667	0.57

**Table 5 materials-15-08092-t005:** Rules matrix.

Sr, No.	N-Hardness (IF)	N-Surface Roughness (µm)	MPCI
1.	S	S	VS
2.	S	M	S
3.	S	L	M
4.	M	S	S
5.	M	M	M
6.	L	S	M
7.	L	M	L
8.	L	L	VL

VS—Very Small; S—Small; M—Medium; L—Large; VL—Very Large.

**Table 6 materials-15-08092-t006:** Optimum results given by TLBO, JAYA, and GA.

Sr. no.	Response	Optimization Method	Orientation (^°^)	Laser Power (KW)	Scanning Speed (mm/s)	Fitness Value
1	Hardness	TLBO	90	136	650	−71.6013 Hv
		JAYA	90	136	650	−71.61 Hv
		GA	90	136	650	−71.60 Hv
2	Surface roughness	TLBO	90	115	650	8.122 µm
		JAYA	90	115	650	8.122 µm
		GA	90	115	650	8.12 µm
3	MPCI	TLBO	90	136	650	0.747244
		JAYA	90	136	650	0.897
		GA	90	135.9	649	0.878

## Data Availability

Not applicable.

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
