# Peer review of "Optimization of Selective Laser Melting Parameter for Invar Material by Using JAYA Algorithm: Comparison with TLBO, GA and JAYA"

_materials, 2022, doi:10.3390/ma15228092_

Round 1

Reviewer 1 Report

Please find my comments in the attached pdf file.

Author Response

Reviewer Response attached.

Reviewer 2 Report

The work is interesting and brings important information to the scientific community, but the authors need to describe better the optimization methods used and also detail how they were used in the problem. In addition, it is necessary to carry out a greater discussion of the results and present better formatting of the graphs and tables.

In the attached file I present some suggestions.

Author Response

Reviewer response attached.

Round 2

Reviewer 1 Report

The authors have done requested revisions from my previous feedback. I have no further technical comment. However, I feel that thorough English language check is required on the manuscript before publication.

Author Response

Authors are very much thankful to the reviewer and as per his suggestions, entire manuscript has been scrolled to rectify grammatical issues.

Reviewer 2 Report

The authors implemented most of the suggestions and with that, I believe that the manuscript can be accepted. Please improve the way you present equation 3.3, maybe if you remove the symbol "*" from the equation it will be better.

Author Response

Authors have corrected the suggested portion of the manuscript. Details are attached.
